# Development of an Implementation Strategy Tailored to Deliver Evidence-Based and Person-Centred Nursing Care for Patients with Community-Acquired Pneumonia: An Intervention Mapping Approach

**DOI:** 10.3390/healthcare12010032

**Published:** 2023-12-22

**Authors:** Signe Eekholm, Karin Samuelson, Gerd Ahlström, Tove Lindhardt

**Affiliations:** 1Department of Health Sciences, Faculty of Medicine, Lund University, Sölvegatan 19, P.O. Box 117, SE-221 00 Lund, Sweden; karin.samuelson@med.lu.se (K.S.); gerd.ahlstrom@med.lu.se (G.A.); 2Department of Internal Medicine, Copenhagen University Hospital Herlev and Gentofte, Gentofte Hospitalsvej 4, 2nd. Floor, DK-2900 Hellerup, Denmark; tove.lindhardt.damsgaard@regionh.dk

**Keywords:** behavioural change, determinants, evidence based, implementation strategy, intervention mapping, implementation science, nursing care, person-centred care

## Abstract

Community-acquired pneumonia is a serious public health problem, and more so in older patients, leading to high morbidity and mortality. However, this problem can be reduced by optimising in-hospital nursing care. Accordingly, this study describes a systematic process of designing and developing a tailored theory- and research-based implementation strategy that supports registered nurses (RNs) in delivering evidence-based and person-centred care for this patient population in a hospital setting. The implementation strategy was developed by completing the six steps of the Intervention Mapping framework: (1) developing a logic model of the problem and (2) a logic model of change by defining performance and change objectives, (3) designing implementation strategy interventions by selecting theory-based change methods, (4) planning the interventions and producing materials through a co-design approach, (5) developing a structured plan for adoption, maintenance and implementation and (6) developing an evaluation plan. This method can serve as a guide to (1) target behavioural and environmental barriers hindering the delivery of nursing care in local clinical practice, (2) support evidence uptake, (3) support RNs in the delivery of nursing care according to individual patient needs and thereby (4) optimise health-related patient outcomes.

## 1. Introduction

Community-acquired pneumonia (CAP) [1,2] is highly prevalent in older populations and a significant cause of mortality, morbidity, prolonged length of hospitalisation and high readmission rates [3,4]. In Denmark, CAP is estimated to be the fifth most common cause of acute hospitalisation and the most common reason for readmission. The incidence of in-hospital mortality has been reported to be 8–11.5%, with considerable implications and high costs for the healthcare system [5,6,7,8].

Efforts have been made to develop clinical guidelines (CGs) that aid in translating the best existing scientific evidence into clinical practice to support healthcare professionals in making decisions regarding appropriate and effective treatment and care [9,10,11]. However, the lack of translation of CG recommendations into clinical practice is widely acknowledged, and studies continue to report variations in in-hospital practice, including inconsistencies in treatment and care [12,13,14,15]. In particular, nursing care has been reported to be delivered haphazardly, unsystematically and, in worst cases, missing, with fatal consequences for patients [12,13,16,17,18]. Although national and international CGs for CAP have been recognised for their thorough review of diagnostic procedures and choice of antibiotic treatment, they do not emphasise the significance of nursing care interventions. The reason is that the description of nursing care interventions is either not described or described sporadically [19], thus indicating the need for improvement.

To overcome the ‘evidence–practice gap’ and improve the adoption of evidence-based practice, tools, programs and strategies have been developed for adoption in clinical practice. One of these tools is a clinical pathway (CPW) that translates high-quality evidence and CG recommendations into local structures and clinical procedures [20]. The goal of CPW is to ensure the overall safety, efficacy and patient-centred care and treatment. The application of CPWs has previously been reported to increase healthcare professionals’ adherence to CG recommendations, improve appropriate timeliness of care and patient outcomes and reduce complications, length of stay and readmissions [21,22,23,24]. Considering the effectiveness of CPWs and the limited description of nursing care interventions in CGs, our research group and clinical experts previously developed a CPW according to the European Pathway Association criteria [20]. Existing national and international CG recommendations were used in combination with a thorough literature review of nursing care interventions for CAP. In contrast to the CGs, the CPW described in detail the responsibilities and duties of registered nurses (RNs) along with interdisciplinary teams (IDT) in planning, delivering and documenting care among patients in the local context. The purpose is to ensure patients receive evidence-based nursing care (EBNC) according to their individual needs (person-centred care). However, previous studies of the successful implementation of CPWs are poorly reported and do not identify factors that contribute to their implementation [21]. Therefore, to support clinicians in transforming scientific evidence into local contexts and working structures, studies investigating evidence uptake in clinical practice through CPWs are needed.

Implementation science has reached a consensus that successful implementation requires a systematic theory- and evidence-based approach and a strong rationale for design [25,26,27,28,29,30]. Furthermore, clear reporting of the development process is crucial for transparency and replication purposes. This process involves the development of a tailored implementation strategy based on (1) the identification of the problem in clinical practice, (2) the identification of gaps between evidence and routine practice and (3) a thorough analysis of the context in which the implementation will take place. The findings are then used in designing the implementation strategy interventions tailor-made for contextual needs.

However, despite an overwhelming amount of literature on the topic, implementation strategies are often haphazardly designed and poorly specified, and theory is insufficiently applied [30,31,32,33,34]. Guidance regarding how best to select tailored strategies to address local contextual determinants (underlying factors, also referred to as barriers and facilitators that cause or influence behaviours and environmental conditions) for successful implementation is insufficient [25,34,35,36], thus limiting the possibility of replication. Furthermore, few studies focus on designing implementation strategies that target change at the individual, team and environmental levels and are particularly relevant in healthcare settings, where healthcare professionals collaborate across disciplines and organisational borders [37]. Thus, studies using a systematic approach and a clear rationale for the design and development of implementation strategies are needed. They can support clinical practices in improving nursing care for patients with CAP and contribute to knowledge accumulation in implementation science [34].

In this study, we present a systematic process of designing and developing a tailored theory- and research-based implementation strategy to deliver EBNC and person-centred nursing care for older CAP patients in a hospital setting.

## 2. Materials and Methods

### 2.1. Setting and Participants

The study was performed in the respiratory unit of a Danish university hospital serving a local population of 700,000 and providing specialised care to approximately 174,000 patients annually. In this 25-bed unit, specialising in respiratory diseases, approximately 10% of acute patients admitted (≥65 years) were diagnosed with CAP. The staff comprised 40 employees, including 20 RNs, eight licensed practical nurses (LPNs), one clinical nurse specialist with a master’s degree and four to five physicians (specialised in respiratory and infectious diseases). Physicians were on call and performed patient rounds daily along with two physiotherapists (responsible for assessing patients’ physical performance, conducting ambulation plans, and delivering specialised respiratory treatment). A head nurse, nurse manager (NM) and two assistant NMs led the unit and were responsible for department operations, interdisciplinary coordination and patient flow.

In standard, nursing care was delivered by the team of one RN and one LPN to approximately 6–8 patients. The RN functioned as a team leader and was responsible for cooperating with the interdisciplinary team of physicians, physiotherapists, etc., according to the patient’s needs for treatment and care. The physicians, RNs, LPNs, NM, assistant NMs and physiotherapists work together to create patient plans at daily interdisciplinary meetings and during patient rounds in cooperation with the patient.

The implementation strategy aims to involve all employees, though the main focus was on the RNs. The development process was carried out in cooperation with NMs (n = 4), a clinical nurse specialist and a purposive group of RNs, LPNs, physicians and physiotherapists (n = 20). Furthermore, eight patients were selected consecutively through purposive sampling to test the implementation strategy materials.

### 2.2. The Intervention Mapping Framework

The Intervention Mapping (IM) framework [38] was applied to design and develop the implementation strategy systematically. The IM is a systematic six-step approach that guides the planning, development, implementation and evaluation of implementation strategies targeting behavioural change. It is an iterative process from the recognition of the problem to the development of an evaluation plan. Table 1 describes all six steps in detail. Completing the tasks in the six steps leads to a tailored, systematic implementation strategy consisting of theory- and research-based interventions and an implementation and evaluation plan [38].

## 3. Development of an Implementation Strategy

In the next section, the development of the implementation strategy will be described by presenting each IM step, followed by a description of the results as the result of each IM step guides the construction of the following step.

### 3.1. Step 1: Logic Model of the Problem (Needs Assessment)

#### 3.1.1. Method

The purpose of this step is to conduct the needs assessment. This is achieved by developing a logic model of the problem, describing (1) the problem in clinical practice, (2) behavioural and environmental factors that contributes to the problem and (3) determinants that cause or influence behavioural and environmental factors. To be able to identify the problem and assess the quality of nursing care, we identified EBNC interventions for CAP patients through the development of a CPW (described previously). The problem in clinical practice (quality of nursing care) was determined by conducting a descriptive cross-sectional study (reported elsewhere [19]). The study used structured participant observations, individual ad hoc interviews with the patients and healthcare professionals and audits of patient records. The aim was to assess the gaps between the recommended EBNC interventions (described in a CPW, see Appendix A) and the current clinical practice for CAP patients. Behavioural and environmental factors and determinants were identified at the individual, team (interdisciplinary) and environmental levels, influencing nursing care in clinical practice. We conducted an ethnographic study (reported elsewhere [39]) using focus group interviews with RNs, LPNs and physicians, field observations and individual follow-up interviews with the RNs.

#### 3.1.2. Results

Figure 1 illustrates the needs assessment in a partial logic model of the problem. 

##### Evidence-Based Nursing Care for CAP Patients and the Problem

Through the review of nursing care interventions, we identified and focused on the following: nutritional support, fluid therapy, oral care, ambulation, airway clearance and oxygen therapy (described in detail in Appendix A). Findings from a descriptive observational study [19] revealed that of all the recommended nursing care interventions, only oxygen therapy was delivered systematically according to individual patient needs. Oral care, fluid therapy, nutritional support and ambulation were either not performed or were delivered inadequately [19]. These findings were defined as the ‘problem’ in the logic model of a problem (Figure 1).

##### Behavioural and Environmental Factors and Determinants Influencing Nursing Care

The ethnographic study [39] revealed multiple behavioural factors among RNs, the team, management and environment contributing to the inadequate and insufficient nursing care. Figure 1 presents primary behavioural and environment factors and the underlying determinants limiting RNs delivery of recommended nursing care. 

### 3.2. Step 2: Logic Model of Change

#### 3.2.1. Method

A logic model of change was developed by determining who and what needs to change at the individual, team, management, and environmental levels to address the problem and thus achieve a successful implementation. The implementation goal is to provide CAP patients with oral care, fluids, nutrition, ambulation, airway clearance and oxygen therapy systematically according to evidence-based recommendations and their individual needs. To reach this goal, the logic model of change was converted by defining the target behaviour of RNs (individual level), interdisciplinary team (team level) and NMs (management level) and determining non-behavioural environmental factors and the related determinants. Thereafter, we created a matrix by linking the targeted behavioural and non-behavioural factors (performance objectives) with the determinants (identified and categorised according to the Theoretical Domains Framework [40]) and related them to change objectives, specifying what needs to be changed.

#### 3.2.2. Results

The matrix visualised what needed to be addressed in the implementation strategy to achieve the implementation goal. Determinants that needed to be targeted were as follows: attention, beliefs about consequences (acknowledgement), knowledge, skills, beliefs about capabilities (self-efficacy), social influence and environmental context and resources. The performance objectives that were linked to determinants and related change objectives, and that were required to achieve the change in clinical practice, were tailored to RNs (individual level), the team, management, and the environment. A detailed description of identified performance objectives is presented in Table 2.

### 3.3. Step 3: Programme Design

#### 3.3.1. Method

In following, we designed the theory-based implementation strategy by matching the determinants at the individual, team, management and environmental levels with theoretically informed change methods. These change methods were then converted into practical applications, specifying practical ways to deliver the interventions.

#### 3.3.2. Results

To ensure the appropriate theory-based change methods, we applied the Behaviour Change Technique Taxonomy [41], supplemented to the IM taxonomy of behaviour change methods [42]. For example, to target the determinant ‘knowledge’ and increase RNs‘ understanding of professional roles, goals and tasks, the taxonomy of behaviour change methods [42] guided us to select the theories of information processing and communication–persuasion matrix. Those theories informed us to use methods as information, elaboration, advanced organisers and persuasive communication techniques to increase RNs’ knowledge and understanding. Then, all the methods were converted into practical applications describing the practical way to carry out the method in the local context. For example, the method ‘guided practice’ (informed by social cognitive theory [43]) was converted into bedside training.

Thereby, the product of this step was a tailored theory-based implementation strategy that aimed to support RNs in the delivery of EBNC according to individual patient needs. This strategy also included multiple implementation interventions targeting behavioural and environmental determinants in local clinical practice. 

### 3.4. Step 4: Programme Production

#### 3.4.1. Method

In an iterative process, the implementation strategy structure and its interventions were tailor-made and refined according to local contextual needs and preferences. Furthermore, we designed, tested and adjusted materials supporting the adoption of the implementation interventions. This step was achieved through a co-design approach [44] involving managers (n = 4), a group of healthcare professionals (n = 20: RNs, LPNs, physicians, physiotherapists and a clinical nurse specialist) and a group of patients (n = 8).

#### 3.4.2. Results

The results of the logic models were presented to the managers and healthcare professionals. The models represent the problem, behavioural and environmental factors and determinants that influence RNs’ ability to deliver EBNC according to patients’ individual needs. The purpose was to increase their knowledge, understanding and motivation for implementation. We also presented the EBNC interventions described in a CPW and the implementation strategy interventions. The participants were encouraged to reflect on and give feedback on the execution of the implementation strategy and discuss the practical delivery of the interventions at two group meetings. The group perspectives and feedback were incorporated into the implementation strategy and in a practical plan for delivery. Implementation intervention materials were tested and adjusted until a consensus was reached over several iterations with all the participants. Table 3 presents an overview of the implementation strategy interventions and materials. A timeframe of 6 months was found reasonable to execute the implementation strategy.

### 3.5. Step 5: Programme Implementation Plan

#### 3.5.1. Method

A project organisation was established, including elaboration of activities and responsibilities for the involved to ensure the adoption, implementation and maintenance of the implementation strategy. The project organisation consisted of (1) a steering group, (2) project managers, (3) implementers and (4) key persons. The steering group included researchers with expert knowledge of implementation, the head nurse and the NM in the targeted unit. The steering group had the overall responsibility to facilitate the adoption of the implementation strategy in the respiratory unit. The project managers were a PhD student (first author) and a senior researcher (last author). Their responsibility was to facilitate the whole implementation process. The implementers were the unit NM, two assistant NMs and a clinical nurse specialist, with responsibility to execute and maintain the implementation strategy interventions. The key persons included four RNs, whose responsibility was to facilitate and support the delivery of nursing care interventions for CAP patients among their colleagues.

#### 3.5.2. Results

The project organisation was presented with the logic model of problem and implementation strategy. They agreed to the adaptation with allocating resources to deliver the implementation strategy. Together with the project organisation, we developed a detailed plan clarifying who does what, where and when to facilitate and support the adoption and maintenance of the interventions. This plan included a weekly meeting with the project organisation (project managers, implementers and the key persons) to evaluate the plan, receive advice and recommendations, gather feedback on the implementers’ performance and, if necessary, adjust the plan and weekly activities according to contextual needs and preferences. The implementers and key persons underwent a training programme on how to execute the implementation strategy interventions.

### 3.6. Step 6: Evaluation Plan

#### 3.6.1. Method

The final step was to develop an evaluation plan to evaluate the implementation process outcomes such as acceptability, adoption, appropriateness, costs, feasibility, fidelity, penetration and sustainability. The evaluation plan was developed by use of the taxonomy of implementation outcomes developed by Proctor et al. [45] and included a detailed plan for data collection.

#### 3.6.2. Result

The data collected for the assessment of the implementation process outcomes included (1) observations (three times a week for 6 months) including ad hoc interviews (individual and in a group with RNs, the team and the management), (2) focus group interviews with the RNs and the team (n = 8) before and after the intervention period, (3) registration of frequency and execution of the implementation interventions (daily registration), (4) audit of electronic patient journals (once a week for 6 months) to assess the quality of nursing care plans (guided by recommended nursing care interventions described in a CPW) and (5) individual interviews with patients (once a week for 6 months) to assess their experience of receiving EBNC according to their individual needs.

The data were planned to be assessed and presented at weekly meetings to facilitate and motivate implementers to adopt and maintain their commitment. Furthermore, the results were presented to the RNs, the team and the managers.

## 4. Discussion

This study presents the design and development process of a tailored theory- and research-based implementation strategy aimed at supporting RNs in the delivery of EBNC according to patients’ individual needs. This implementation strategy is guided by the IM framework and consisted of multiple implementation interventions targeting behavioural and environmental determinants in clinical practice. These interventions are expected to support the implementation of the innovation (evidence-based recommendations for CAP patients described in a CPW). The study process and end-product are unique in that they address one of the major challenges in a healthcare setting, namely missed nursing care. This issue is not only a common concern among CAP patients but also a general worldwide challenge in healthcare settings [13].

The majority of older patients with CAP admitted to acute care are often frail, and thus, the responsibility for their physical and psychosocial well-being is in the hands of healthcare professionals. According to the IM framework, steps 1 and 2 in the IM framework reveal that CAP patients’ individual needs for fundamental care are not adequately met. Similarly to our study, previous research reveals this as a difficult task [15,16,17,18,19]. This is a global phenomenon with an estimated prevalence of 55–98% in acute care hospitals [16,17,18,46,47]. Low adherence to guidelines for nursing care may have fatal consequences on CAP patients, thus constituting a threat to their safety [48,49,50,51,52,53,54]. In past decades, researchers and clinicians have endeavoured to address the major public health challenge of CAP and promote the uptake of evidence in clinical practice. Moreover, numerous policy strategies have been developed, emphasising the importance of timely and evidence-based treatment and care to maximise clinical efficiency. However, the majority of these strategies focus on effective diagnostic procedures and medical treatment, whereas nursing care is a neglected area [55]. Moreover, little is known about the reasons for suboptimal nursing care, including care for CAP patients. Therefore, a broader understanding of the context of nursing care for hospitalised patients is necessary.

In this study, the logic models (IM steps 1 and 2) revealed the need for a broader understanding of local clinical practice in a hospital setting to understand the factors that influence RNs and their delivery of nursing care. The needs assessment showed that, in addition to the RNs, the IDT and the management were important target groups to address with interventions as their behaviours strongly influenced RNs’ delivery of nursing care. Additionally, the organisational environment constituted a significant barrier to the delivery of nursing care. Hence, the logic models (steps 1 and 2) emphasised that multiple tailored interventions were required to achieve the implementation goal and ensure successful execution. Our design and development process has been carried out according to the growing evidence indicating that implementation strategies based on multiple interventions tailored to target determinants at multiple levels are more effective in implementing change and improving professional practice than single intervention strategies [31,56,57]. However, a systematic review by Colquhoun [37] found that few studies have developed implementation strategies targeting change at multiple levels. Most studies focus on individual or environmental determinants. Considering a healthcare setting where RNs collaborate with multiple other disciplines and across the organisation, the assessment of determinants at multiple levels is essential to achieve success and avoid research waste. Moreover, a systematic review by Lewis et al. [58] emphasised the importance of designing strategies for clinical practice that target specific local determinants to avoid the deployment of insufficient or less effective strategies, resulting in a waste of time and resources. Our study presents an example of how to design and develop an implementation strategy that meets the above-mentioned recommendations.

IM step 3 guided us to develop interventions for the implementation strategy, applying systematic use of relevant theory and empirical data. In the design process, we illustrated how empirical data of behavioural and environmental factors and determinants can be matched with theory-based change methods. The systematic use of theory is in line with the recommendations from implementation science, claiming that this rigorous approach drawing on theory and empirical data is a prerequisite for developing a successful implementation strategy, e.g., [27,28,29,30,37,59]. According to Birken et al. [31], this approach facilitates the selection of relevant and appropriate interventions targeting determinants, as well as guides the actual implementation activities. Moreover, Nilsen [60] and Bergström et al. [61] report that the use of theory to develop interventions targeting behavioural change provides guidance not only for developing and tailoring implementation interventions but also for evaluating the implementation process and increasing understanding of the factors that influence implementation outcomes. Despite these recommendations, implementation strategies are often designed unsystematically and do not specify how theories and evidence were used [31]. Moreover, in a systematic review assessing the role of IM in designing disease prevention interventions, Garba et al. [62] found that most IM studies fail to provide details on data collection and analysis, thereby limiting the methodological quality, validity and transferability. In the development of an implementation strategy, we found that the IM framework is helpful in guiding how to combine determinants identified in clinical practice with theory-based interventions. In the subsequent evaluation study, we expect to find that addressing theory-based implementation interventions will facilitate the adoption and successfully support RNs in the delivery of high-quality, EBNC and person-centred nursing care, which is claimed to be in short supply in hospital settings worldwide [16,55]. Consequently, we expect a positive impact on CAP patients’ health-related outcomes.

In steps 4 and 5, we applied a co-design action research approach to prepare the interventions, materials and a detailed plan for the adoption, implementation and maintenance of the strategy [44]. This part ensured that the developed strategy and materials were contextually relevant and tailored to clinical preferences and needs. By incorporating clinical expertise, we obtained immediate feedback from the participants and captured their perspectives on issues that could be addressed immediately. Other researchers have reported that incorporating clinical expertise into the development process is a strength and a prerequisite for a feasible, acceptable and sustainable implementation strategy; it fosters a sense of ownership among implementers and ensures acceptance and successful implementation [25,38,63,64]. Moreover, a systematic review by Kwasnicka [65] reports that the motivation and goal setting that corresponds with participants’ preferences are effective and supportive in changing professional behaviours. Therefore, end-users should be integrated into the early stages of the design and development process [25,35,44].

In step 6, we designed a detailed evaluation plan by using the Implementation Outcomes Framework [45], enabling a continuous evaluation of the implementation process and assessment of outcomes. A systematic review by Wagenaar et al. [66] stated that although the field of implementation science aims to test strategies for optimising the implementation, only a few studies have evaluated and optimised strategies for scaled-up interventions in clinical practice. Most studies have tested strategy effectiveness using randomised controlled trials in which the context is controlled for rather than in a routine clinical practice setting that can be unpredictable and dynamic [66,67]. As stated by Powell et al. [31], future implementation studies should increasingly focus on describing the processes by which strategies exert their effects rather than establishing whether these strategies are effective.

Certain methodological considerations and limitations should be discussed. First, it is noteworthy that as the strategy was tailored to a medical unit in a local university hospital, it may not be transferable to another context without tailored adjustments, especially as the strategy was developed with focus on delivery of nursing care for patients admitted with CAP. Nevertheless, the systematic theory-based approach is transferrable, as it can be used to improve the clinical practice no matter the clinical speciality or the patient group, which is in conjunction with findings from previous research assessing the role of the IM [62,68,69]. Second, the use of IM made advantageous contributions to this study as we achieved to develop a tailored, theory-based strategy and a plan for implementation and evaluation in an iterative co-design process, as recommended when developing a complex strategies in a healthcare setting [25,26,27,28,29,30,37,38,44]. However, it is important to mention that several other frameworks and taxonomies needed to be identified and used in this study to complete the IM steps. For example, when developing the logic model of the problem, we applied the Theoretical Domains Framework to identify behavioural and environmental factors and determinants. Moreover, due to the absence of firm guidance on which interventions should be linked to which change objectives, we facilitated the process by using the Behaviour Change Technique Taxonomy [41]. This approach resulted in the development of a strategy based on multifaceted interventions targeting not only the determinants among the RNs but also among the interdisciplinary team, the management and the environment as recommended and called for in implementation science [37]. Consequently, our strategy can be considered as too comprehensive to carry out in a busy clinical practice and needs to be tested to assess the feasibility. Notably, although implementation science provides us with numerous models and frameworks supporting the development of implementation strategies, the process is complex and requires time and expertise. In this study, all six IM steps were completed in approximately 6 months. Considering the complexity of the process and the length of time used, the framework is challenging for frontline healthcare professionals to apply. Therefore, the design and development of tailored implementation strategies should be carried out by an experienced researcher or implementation expert to increase the possibility of success [63,70].

Despite the complexity of the IM approach, this systematic and stepwise process is highly beneficial and adequate for developing an implementation strategy in clinical practice, in combination with the use of the Theoretical Domains Framework and the Behaviour Change Technique Taxonomy. Through the detailed description of the systematic design process, we expect to have achieved transparency and replicability for the benefit of others in designing theory- and research-based implementation strategies.

## 5. Conclusions

This study describes in detail the design and development of a tailored, theory-driven and research-based implementation strategy aimed to facilitate a clinical context supporting RNs in delivering EBNC and person-centred nursing care for hospital patients with CAP. The thorough report of this systematic approach can serve as a guide for future researchers in developing implementation strategies and selecting interventions that will overcome local determinants, thereby enhancing the possibility of a successful implementation.

## Figures and Tables

**Figure 1 healthcare-12-00032-f001:**
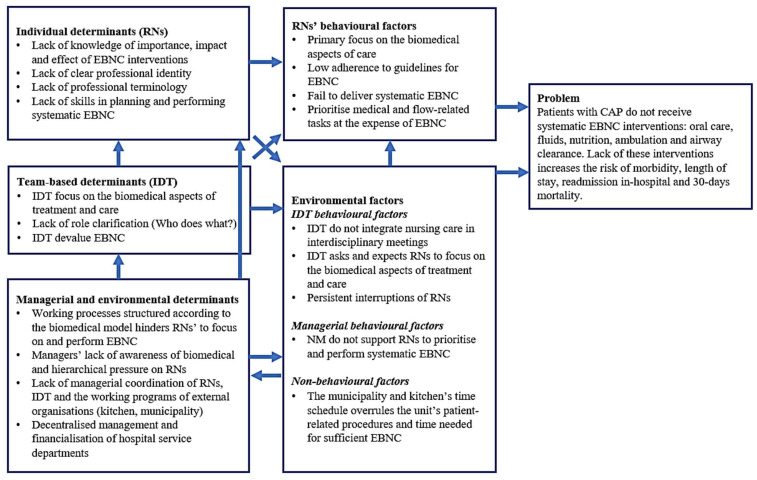
Partial logic model of the problem. CAP: community-acquired pneumonia; EBNC: evidence-based nursing care; IDT: interdisciplinary team; LPNs: licensed practical nurses; NM: nurse manager; RNs: registered nurses.

**Table 1 healthcare-12-00032-t001:** Six Intervention Mapping steps, adopted from Bartholomew et al. [38].

Step	Tasks	Definitions
Logic model of the problem	Conduct a needs assessment to create a logic model of the problem (quality of care) and its underlying factors and determinants	*Determinants*: factors that cause or influence a behaviour or environmental condition
2. Logic model of change	State expected outcomes and performance objectives for behaviour and environment Select determinants for behavioural and environmental outcomes Construct the matrixes of change objectives	*Performance objective*: state the tasks of programme participants or how the environment will be modified *Behavioural and environmental outcomes*: programme outcome statements that the developed implementation strategy seeks to accomplish *Change objective*: determine the required changes to determinants to accomplish performance objectives
3. Programme design	Select theoretically informed change methods Design practical applications to deliver change methods	Theoretical methods: techniques influencing changes in behavioural and environmental determinants Practical applications: delivery of a change method that fits the needs and preferences of the context
4. Programme production	Refine programme structure and organization Design, pre-test and refine materials	
5. Programme implementation plan	Identify programme users (adopters, implementers, maintainers) Design implementation interventions	
6. Evaluation plan	Develop an evaluation plan Select evaluation measures for assessment	

**Table 2 healthcare-12-00032-t002:** Performance objectives at the individual, team, management, and environmental levels needed to be targeted in the implementation strategy. Øverst på formularen.

Individual Level	Team Level	Management Level	Environmental Level
Acknowledge professional role, goals and tasks Use professional terminology Plan, perform, document, evaluate and adjust EBNC and person-centred nursing care as described in a CPW and according to individual patient needs Resist social pressure and cooperate with the IDT, with a focus on own professional role, goals and tasks	Cooperate and support RNs in planning, performing, evaluating and adjusting nursing care Integrate RNs and nursing care into interdisciplinary meetings and patient rounds Eliminate behaviours and interruptions, supporting RNs in the delivery of nursing care	Facilitate RNs to plan, perform, document, evaluate and adjust EBNC and person-centred nursing care for CAP patients Eliminate facilitate individual- or team-based behaviours, supporting the achievement of the implementation goal Initiation of environmental changes supporting the delivery of nursing care.	Reorganisation of the IDT, external services (i.e., central kitchen, municipality visitations) and working processes Reconstruction of the documentation system to support RNs in planning, performing, evaluating and adjusting nursing care.

CAP: community acquired pneumonia; CPW: clinical pathway; EBNC: evidence-based nursing care; IDT: interdisciplinary team; RNs: registered nurses.

**Table 3 healthcare-12-00032-t003:** Overview of the interventions and materials in the implementation strategy.

Individual Level (RNs, LPNs)	Team Level (IDT)	Environmental (NM) Level
Presentations of previous research results of behavioural and environmental factors, determinants and consequences Information of the implementation strategy Lectures and open debate of: ○ the professional role, tasks, goals, and terminology ○ the evidence base of the interventions in the CPW Group training sessions of the practical execution of the EBNC interventions in the local context in cooperation with the team Individual bedside training and supervision of the practical execution of the EBNC interventions Daily reminders and facilitation of EBNC Nudging of EBNC performance Feedback of individual performance and project results	Presentation of previous research results of behavioural and environmental factors, determinants, and consequences Encourage physicians to support RNs Facilitate LPNs to support RNs Daily reminders and facilitation Feedback of team performance and project results	Presentations of previous research results of behavioural and environmental factors, determinants, and consequences Workshop of the facilitating and eliminating strategies to change staff behaviours and environmental conditions Planning, discussion, and decision making Feedback of management performance and project results
**Materials**		
Educational materials as quizzes, videos, etc., regarding EBNC interventions, described in the CPW Newsletters, folders informing about the project Learning contract and the evaluation sheet for the individual bedside training Bedside whiteboard magnets describing the individual patient’s needs for nursing care Daily patient list as reminder for RNs and LPNs regarding EBNC interventions tailored to each patient’s individual needs Pocket cards guiding the interdisciplinary whiteboard meeting and patient rounds	CPW (paper and electronic) Newsletters, folders informing about the project Pocket cards guiding the interdisciplinary whiteboard meeting and patient rounds	Guide for NM of facilitating and eliminating strategies

CPW: clinical pathway; IDT: interdisciplinary team; LPNs: licensed practical nurses; NM: nurse managers; RNs: registered nurses.

## Data Availability

The datasets generated and analysed during the current study are not publicly available due to information that could compromise the privacy of the research participants but are available from the corresponding author upon reasonable request.

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
