# Peer review of "Development of an Implementation Strategy Tailored to Deliver Evidence-Based and Person-Centred Nursing Care for Patients with Community-Acquired Pneumonia: An Intervention Mapping Approach"

_healthcare, 2023, doi:10.3390/healthcare12010032_

Round 1

Reviewer 1 Report

Comments and Suggestions for Authors

-In the stratification of a plan, especially in a pandemic era in which Covid-19 seems to be easing but has now established itself and often takes the form of a nosocomial infection, an internal epidemiological and infectious disease investigation to prevent nosocomial infections such as pneumonia, should always be conducted. Please add references on the topic (doi: 10.21037/atm-20-3324; doi:10.1186/s13756-020-00875-7; doi:10.3390/pathogens1205064). In this case the nurse could have an important role in the preventive stratification of a risk of nosocomial infection. Please explain this aspect further in the manuscript.

-The nurse could also assist patients in providing informed consent for necessary treatments and C-PAP. This both in a hospital and in a home context care. Please add new references on the topic (e.g. doi:10.3390/healthcare11121793; doi:10.3390/vaccines9050429; doi:10.1177/0969733012468).

-The roles of the figures who perform nursing activities (head nurse, nurse manager (NM) and 106 two assistant NMs headed the unit) are described very briefly or not at all. In each single step, the individual roles should be specified.

-The limits are very generic and it is not clear how the entire process described can be rectified or improved, as well as applied in hospital settings other than the one reported in the article. The authors should clarify this aspect in the Discussion section, also making a comparison with the scientific literature cited.

 Minor revisions:

-please make the tables overwritable, and not as figures.

-I don't see reference 74 inserted correctly in the text.

Comments on the Quality of English Language

The article must be submitted entirely in the British language or entirely in the American language.

Author Response

For research article: Development of an Implementation Strategy Tailored to Deliver Evidence-Based and Person -Centered Nursing Care for Patients with Community Acquired Pneumonia: An Intervention Mapping Approach.

Manuscript ID: healthcare-2738513

Response to Reviewer 1 Comments

REVIEWER COMMENTS

AUTHORS RESPONSES

Comments and Suggestions for Authors

In the stratification of a plan, especially in a pandemic era in which Covid-19 seems to be easing but has now established itself and often takes the form of a nosocomial infection, an internal epidemiological and infectious disease investigation to prevent nosocomial infections such as pneumonia, should always be conducted. Please add references on the topic (doi: 10.21037/atm-20-3324; doi:10.1186/s13756-020-00875-7; doi:10.3390/pathogens1205064). In this case the nurse could have an important role in the preventive stratification of a risk of nosocomial infection. Please explain this aspect further in the manuscript.

Thank you very much for taking the time to review this manuscript.

We are very grateful for all your comments and suggestions that we have considered thoroughly. We have made revisions throughout the manuscript, and we hope you will find them satisfying.

Please find the detailed responses below and the corresponding revisions/corrections highlighted/in track changes in the re-submitted files.

We agree on your comment on nurse’s role in preventing stratification of a risk of nosocomial infection. However, considering the study aim and scope involves nursing care for patients with a community acquired respiratory infection, we find the suggested intervention and references difficult to include in this manuscript.

The nurse could also assist patients in providing informed consent for necessary treatments and C-PAP. This both in a hospital and in a home context care. Please add new references on the topic (e.g. doi:10.3390/healthcare11121793; doi:10.3390/vaccines9050429; doi:10.1177/0969733012468).

Thank you for this suggestion that we have considered. We have thoroughly read the suggested references and considered the content, but do not find that the references support the scope and aim of the study. Moreover, the implementation strategy was developed for a general unit for respiratory diseases where C-PAP treatment is not applied by RNs, hence we have not included C-PAP treatment as part of the strategy.

The roles of the figures who perform nursing activities (head nurse, nurse manager (NM) and two assistant NMs headed the unit) are described very briefly or not at all. In each single step, the individual roles should be specified.

We are not sure we understand what you mean here. Are you referring to any of the figures in the paper? The nursing activities recommended to be delivered by RNs are described under the paragraph ’Evidence-based nursing care for CAP patients and the problem’ on line 173-181, and on line 214-216.

The planned activities for the managers related to the implementation are described under the section ’Programme implementation plan’ on line 315-329 and an overview of activities are presented in Table 3 on line 345-346

Considering your suggestion, we have expanded description of mangers role under the section ‘Setting and participants’ on line 115-117.

We hope this will address your request satisfactorily.

The limits are very generic and it is not clear how the entire process described can be rectified or improved, as well as applied in hospital settings other than the one reported in the article. The authors should clarify this aspect in the Discussion section, also making a comparison with the scientific literature cited.

Thank you for this comment that we appreciate. We agree on that the limitation section could be expanded. We hope our revisions on line 479-501 are satisfying

Minor revisions:

 -please make the tables overwritable, and not as figures.

Thank you for your suggestion, we have revised the tables according to your suggestion

I don't see reference 74 inserted correctly in the text.

We appreciate that you have noticed this error. We have revised and inserted reference 74 (now ref nr. 76)

correctly on line 545. We needed also to revise ref. 72 (now ref. nr. 75) on line 539

Comments on the Quality of English Language

The article must be submitted entirely in the British language or entirely in the American language.

Thank you for your suggestion. The manuscript has been revised by the Language Editing Service Scribendi and revisions are made according to British English

Reviewer 2 Report

Comments and Suggestions for Authors

Appreciate the extensive work on the manuscript.  Interesting manuscript and agree 100% in integrating the nursing in the care of CAP patients.  I found some sections hard to follow and repetitive.  I have made some revision and few English terminology suggestions for your consideration.  I hope my comments/suggestion are easy to follow.

Good luck.

------------------

Manuscript ID: healthcare-2738513

Title: Development of an Implementation Strategy Tailored to Deliver Evidence-Based and Person -Centered Nursing Care for Patients with Community Acquired Pneumonia: An Intervention Mapping Approach

General Comments: Appreciate the extensive time and effort preparing the manuscript.  The manuscript does not have major English language editorial issues; however, it is very difficult to follow and comprehend due to repetitive sections and phrases/terminology that overlap.  Some sections were easy to make comments and suggestions, some I could not since I was not sure of the authors intention.  I hope my comments are clear and will be well received. 

Few general comments:

Consider replacing person-centered with patient-centered, this is the common terminology used.

Terms implementation and evidence are very repetitive throughout the manuscript and sometimes cause confusion.   Not sure if other terms can be used instead.  I will try my best to make some suggestions.

Abbreviations were a little difficult to follow, but I got through. 

Considering the term “determinants,” I wonder if it will be better to use barriers or factors in some sections.

ABSTRACT:

Line #14 – Consider revising the first sentence to get attention of the readers.  Start with: Community -acquired pneumonia (CAP) is a serious public health problem and more so in older individuals leading to high morbidity and mortality.     

Line 18 – Just end the sentence with: for this patient population in a hospital setting.  Delete “older patients with CAP.”

Line #25 – Here is the first section for the term “determinants” could be switched to “barriers” hindering the delivery of …

INTRODUCTION:
Line #32 – Line #37 - Revise and simplify to: Community-acquired pneumonia (CAP) is highly prevalent in older population and estimated to be the fifth most common cause of acute hospitalization in Denmark.  The incidence of in-hospital mortality has been reported to be 8-11.5% with a high morbidity, mortality, prolonged hospitalization, a high readmission rates, and even a burden to the healthcare costs.

Line #47 – I would suggest changing “the importance” to “the significance or the value” of nursing.

Line #50 – Line #52 – Consider changing “tools” to “programs” and strategies have been developed for adoption in clinical practice.  This eliminates the term “evidence” repeated three times.  The in line #52 say one of these programs is CPW.

Line #54 – Line #55 – Revise the sentence to:  The goal of CPW is to ensure the overall safety, efficacy, and patient-center care treatments.

Line #62 – Delete “patients” at the end of the sentence. 

Line #63 – Consider replacing “tasks” with “duties” of RNs along with IDTs in planning, delivering and documenting care among patients.  No need to say CAP, by now we know we are focusing on CAP.

Line #66 – Replace “person” with “patient-centered” care.     

Line #67- I think here evidence can be replaced with Proof of the successful implementation of CPWs is poorly reported and does not identify the factors that contribute to their execution or application.

Line #78 – Line #79 – delete this sentence, not necessary just extra wording and does not add to the context. 

Line #95 – Here again we have Implementation strategy and implement.  I wonder if you can change to: implementation strategy to incorporate or integrate EBNC and patient-centered nursing care….

MATERIALS & METHODS

Line #100 – Consider revising to: providing specialized care to approximately….

Line #106 – This is just a comment.  You mention physiotherapist, since you mentioned the study was conducted in a specialized ID and respiratory facility, not sure why not respiratory therapists?  I do realize that PT’s can perform multiple activities.

Line #109 – Line #111- Consider revising this sentence to eliminate unnecessary wording.  Each team provided direct nursing care to approximately 6-8 patients and one charge nurse as a leader for care coordination.   

Line #118 -Delete “involving patient care”, not necessary.

DEVELOPMENT of an IMPLEMENTATION STRATEGY

Line #130 – in this short paragraph, you have repeated IM three times.   I think under each IM framework you can just define/describe the methods and the results.  Not sure if you need to repeat the title and say in this step, in the next step, or in the final step.  Just a comment.

Line #138 – Delete “for CAP patients” again it is repetitive.

Not sure why we went on to a different paragraph starting with Line #143 and talk about Third.  I think this section will require some adjustment and should be combined to one paragraph (Line #135 – Line #151).

Line #157 – The title (line #156) already has “Cap patients”, therefore delete “for CAP patients”.  Also, here you have interventions back-to-back.  So, maybe consider deleting the second one and say, we identified and focused on the following: nutritional support, fluid therapy, oral care, ambulation (delete the mobilization to avoid duplicate) and sputum mobilization or airway clearance and oxygen therapy. 

Line #162 – Line #163 – Delete interventions at the beginning of the sentence, just start the sentence with: Oral care, fluid therapy…. were either not performed or were delivered inadequately.  Delete “undone or haphazardly.” 

Line #167 – Delete “that”, not needed.  Here also consider the following revision to clean up and help the flow:  The ethnographic study revealed multiple behavioral factors among RNs, the entire team, management, and the environment responsible for the inadequate and insufficient care. 

Line #169 – Line #184 - Consider eliminating extra repetitive sentences and say: Figure 1 presents the primary focus and underlying influence(s) to RNs behaviors limiting their delivery of recommended nursing care.   In addition, figure 2 (additional file) presents the overall overview of behavioral factors and determinants.  Not sure why you have repeated Figure 1 three times.  Just combine the two paragraphs starting line #167 – Line #184.  Then place the figure just after line #184. 

Line #180 is a repeat from Line #170 reporting on the biomedical aspects limiting nursing care.

Line #173 – Just abbreviate IDT, not sure why you have the team (interdisciplinary), you have used the abbreviation multiple times.

 Line #190 – Change “mobilization” to “ambulation”,

Line 200 – Line 220 – You have listed information here and reference table #2.  May consider eliminating some duplication.

Line #228 – Revise to delete extra words: to ensure the appropriate theory-based change methods, we applied the BCTT design or guide.  Delete the extra words at the end.

Line #230 – Line #239 – Not sure if can be simplified for the knowledge information. 

Line #242 – Change “aims” to “aimed”. Also Line #143 change “includes” to “included” for proper past tense.

Line #256 – No need to say “we held an event.  Just start the paragraph with: The results of the logic models were presented to the managers and ….

Line #257 and Line #258 – Correct “represented” and “influenced”, again past tense.

Line #266 – Delete “below” not needed. 

Line #271 – Correct includes to “included”.

Line #277 – Revise to jut in the respiratory unit or ward. Also, delete (before “The.”

Line #281 – This sentence is repeat of the Line#279 for implementation and execution as a responsibility, please delete.  Or, it sounds like there was an overlap of responsibilities!!!

Line #286 – Line #289 -This is a suggestion for you to consider:  The project organization was presented with the logic model of problem (environmental and behavioral) and implementation strategy.  They agreed to the adaptation with allocating resources to deliver the implementation strategy.  Delete (economic and timely resources).   Delete the repeated “the problem” in the parenthesis.

Line #302 – Line #304 – I think part of the results should be under the Method to describe the plan. Under method consider: The final step was to evaluate the implementation process outcomes such as acceptability, adaptation, … The evaluation plan was developed based on guidance from the stated ….by Proctor et al.

Line #305 – Line # 320 - Under results now start with: A detailed data collection for the implementation process, outcomes including acceptability, adoption, feasibility, cost, fidelity, and sustainability was developed.  The data collected included: 1) observations……continue until the end of 320 with the except of Line #314 – Delete “on admitted CAP patients” sine you already say electronic patient journals, should this even be patient’s electronic medical records?

DISCUSSION

Line #325 – Delete the second “implementation” and just say consisted on multiple interventions.

Line #332 – Line #338 -What do you mean by “acutely admitted patients???? Consider expanding or clarifying, since later you mention in acute care hospitals missing care is estimated…

Maybe starting with: patients admitted to acute care facility are often ill and frail and are provided care in an inpatient setting (physical and psychological).  According to the IM framework, steps I and 2 fundamental care of pneumonia patients are not adequately met.  This is like our study and previous research revealing this as a difficult task.  This is a global phenomenon with an estimated prevalence of 55-98% in acute care hospitals.

Line #338 – Line #339 – Delete the sentence: Low adherence…. does not necessary always cause death.   The care is just not optimal.

Line #356 – This implementation should be changed to execution.  Sine you just mentioned achieve implementation goal and then you execute the project.

Line #362 – Consider changing to: and across the organization. 

Line #366 – The term powerful or vigorous may be better here instead of “potent”.

Line #391 – Change person to patient-centered care

Line #399 – Change “handled” to “addressed”.

Line #405 – Not sure if “consumers” may be a better fit than “end-users”.

TABLES

TABLE #1 – I would suggest spelling out IM at the title since we do not have additional abbreviations listed for this table.

#2 Logic Model of Change – Third bullet under definition, sounds confusing; can you just change it to: Change Objectives: determine the required changes to barriers (as suggested a terminology change previously) to accomplish performance objectives.

TABLE #2 – Very difficult to follow, congested and some wording/terminology can be changed.

Not sure what is the purpose of having “Implementation strategy outcome: Older patients ….in the top row of the table!!! Consider deleting.

Delete “Abbreviations in table 2; usually the abbreviations are just listed. 

Under attention may be more appropriate to use: increase awareness on own professional….

Under individual level, own professional goal is repeated for all titles, not sure if I understand the purpose.

TABLE #3 – Same as above: Delete “abbreviations in table 2, just list the abbreviations.

Under Individual level (first column)

“Education” or “workshops” vs. lectures on professional roles….

Not sure if “Nudging” term is appropriate here.  Have not seen in used in the past for many reviews I have done.  So, maybe change the term.

I believe term encourage will be better for facilitate in the second column; i.e., encourage physicians and or LPNs to support RNs. 

Under environmental Level (third column)

What do you mean by “lectures of the facilitating and eliminating strategies”?

Under Materials (second row)

Delete “Forms”, just say CPW (paper or electronic)

Not sure if you need to be specific about six nursing interventions since

FIGURE #1.

You have figure on reference on page 5 in three lines #164, 172, and immediately #174). Can this be eliminated and references at just two or even at the end of the section? 

In Environmental determinants

Not sure if the first bullet point is clear,

What do you mean by decentralized management and financialization of hospital services?

In the problem section

Change mobilization to ambulation as mentioned previously.

Non-behavioral factors

What do you mean by the municipality and kitchen’s time schedule…….If it is with regards to nutrition, it may be more appropriate to say get nutritional consult. 

Comments on the Quality of English Language

Did not see extensive English Language issues.  However, as mentioned above, I have made several suggestions (including English Language) for the authors for consideration.  The supplemental documents are extensive.  

Author Response

For research article: Development of an Implementation Strategy Tailored to Deliver Evidence-Based and Person -Centered Nursing Care for Patients with Community Acquired Pneumonia: An Intervention Mapping Approach.

Manuscript ID: healthcare-2738513

Response to Reviewer 2 Comments

REVIEWER COMMENTS

AUTHORS RESPONSE

General Comments: Appreciate the extensive time and effort preparing the manuscript.  The manuscript does not have major English language editorial issues; however, it is very difficult to follow and comprehend due to repetitive sections and phrases/terminology that overlap.  Some sections were easy to make comments and suggestions, some I could not since I was not sure of the authors intention.  I hope my comments are clear and will be well received. 

Thank you for a very thorough revision and highly useful comments. We are very grateful for all your comments and suggestions, and we hope you will find our revisions satisfying.

Please find the detailed responses below and the corresponding revisions/corrections highlighted/in track changes in the re-submitted files.

Few general comments:

Consider replacing person-centered with patient-centered, this is the common terminology used.

Thank you for this suggestion which we have considered, but we, respectfully, disagree.Person-centred care recognises that optimal care is centred on a person’s needs, preferences and values in the context of their lives, rather than only their symptoms and diagnosis as a hospital patient.

It is a shift from viewing the patient as a passive target for the activities of a healthcare system to another view (person-centered) where the person with an illness is an active part in his or her care and decision-making. We have in this study intentionally focused on person-centered care, and therefore we prefer to keep this term.

Terms implementation and evidence are very repetitive throughout the manuscript and sometimes cause confusion.   Not sure if other terms can be used instead.  I will try my best to make some suggestions.

We have revised the wording according to your suggestion trough out the manuscript and have deleted repetitive words where it is appropriate:

Line 57, 74 and 85.

Abbreviations were a little difficult to follow, but I got through. 

We have considered your suggestion and erased abbreviations BCT and TDF

Considering the term “determinants,” I wonder if it will be better to use barriers or factors in some sections.

Thank you for your consideration. We have replaced the term ‘determinant’ with barrier where appropriate (on line 26-27, 91, 521 and in Table 3). However, as the IM approach uses the term ‘determinant’ in development process of the logic models, we prefer to keep the term where necessary.

We have also revised the definition of determinant in Table 1 and clarified the term on line 91-92 and 151-152

ABSTRACT:

Line #14 – Consider revising the first sentence to get attention of the readers.  Start with: Community -acquired pneumonia (CAP) is a serious public health problem and more so in older individuals leading to high morbidity and mortality.     

Thank you for this suggestion, we have revised the sentence on line 34-38

Line 18 – Just end the sentence with: for this patient population in a hospital setting.  Delete “older patients with CAP.”

We have revised the sentence accordingly on line 19-20, at page 1

Line #25 – Here is the first section for the term “determinants” could be switched to “barriers” hindering the delivery of …

Thank you for this suggestion, we have revised the sentence on line 26-27, at page 1

INTRODUCTION:
Line #32 – Line #37 - Revise and simplify to: Community-acquired pneumonia (CAP) is highly prevalent in older population and estimated to be the fifth most common cause of acute hospitalization in Denmark.  The incidence of in-hospital mortality has been reported to be 8-11.5% with a high morbidity, mortality, prolonged hospitalization, a high readmission rates, and even a burden to the healthcare costs.

Thank you for this suggestion, which we have discussed. We can see that there was a need for simplification, however we have chosen to revise the sentences differently than suggested. Line 34-40

Line #47 – I would suggest changing “the importance” to “the significance or the value” of nursing.

We have revised the sentence accordingly on line 51, at page 2

Line #50 – Line #52 – Consider changing “tools” to “programs” and strategies have been developed for adoption in clinical practice.  This eliminates the term “evidence” repeated three times.  The in line #52 say one of these programs is CPW.

Thank you for this suggestion, which we have discussed. We have included the word programs, however, as the CPW is defined as a tool we cannot replace it with the program. Although, we have revised the sentence to avoid repeat of the term “evidence”.

Revision on line 55-56, at page 2

Line #54 – Line #55 – Revise the sentence to:  The goal of CPW is to ensure the overall safety, efficacy, and patient-center care treatments.

We have revised the sentence accordingly on line 59-61, at page 2

Line #62 – Delete “patients” at the end of the sentence. 

We have revised the sentence accordingly on line 68, at page 2

Line #63 – Consider replacing “tasks” with “duties” of RNs along with IDTs in planning, delivering and documenting care among patients.  No need to say CAP, by now we know we are focusing on CAP.

Thank you for this suggestion. We have replaced the term “tasks” with “duties” and revised the sentence on line 69, at page 2

Line #66 – Replace “person” with “patient-centered” care.     

As described previously, we prefer to keep the term ’person-centered’

Line #67- I think here evidence can be replaced with Proof of the successful implementation of CPWs is poorly reported and does not identify the factors that contribute to their execution or application.

Thank you for this suggestion, which we have discussed. Instead of term ‘proof’, we have replaced it with ‘studies’, as it in fact is the study process and methodology that is poorly reported. Line 74, at page 2

Line #78 – Line #79 – delete this sentence, not necessary just extra wording and does not add to the context. 

Thank you for this suggestion. We agree and have deleted the sentence on line 84-85

Line #95 – Here again we have Implementation strategy and implement.  I wonder if you can change to: implementation strategy to incorporate or integrate EBNC and patient-centered nursing care….

We replaced the term ‘implementation’ with ‘deliver’ as also described in the title. Line 102, at page 3

MATERIALS & METHODS

Line #100 – Consider revising to: providing specialized care to approximately….

We have revised the sentence accordingly on line 107, at page 3

Line #106 – This is just a comment.  You mention physiotherapist, since you mentioned the study was conducted in a specialized ID and respiratory facility, not sure why not respiratory therapists?  I do realize that PT’s can perform multiple activities.

Thank you for your comment. In this particular hospital unit, physiotherapists have a specialized skills and responsibility for ambulation of patients, whereas RNs are specialized in delivering respiratory treatment. However, we have added explanation of physiotherapists function in the study on line 113-115, at page 18

Line #109 – Line #111- Consider revising this sentence to eliminate unnecessary wording.  Each team provided direct nursing care to approximately 6-8 patients and one charge nurse as a leader for care coordination.   

We agree on that the sentence needs a revision. Revision has been made on line 118-123, page 3

Line #118 -Delete “involving patient care”, not necessary.

Revised accordingly on line 131-132, at page 3

DEVELOPMENT of an IMPLEMENTATION STRATEGY

Line #130 – in this short paragraph, you have repeated IM three times.   I think under each IM framework you can just define/describe the methods and the results.  Not sure if you need to repeat the title and say in this step, in the next step, or in the final step.  Just a comment.

We have revised the sentence on line 145, at page 3.

We have revised the text according to your suggestion about the repetitions:

line 148, 211, 262, 291, 315, 352

Line #138 – Delete “for CAP patients” again it is repetitive.

Revised accordingly on line 154-155

Not sure why we went on to a different paragraph starting with Line #143 and talk about Third.  I think this section will require some adjustment and should be combined to one paragraph (Line #135 – Line #151).

We have considered your suggestion and made revisions on line 152-155

Line #157 – The title (line #156) already has “Cap patients”, therefore delete “for CAP patients”.  Also, here you have interventions back-to-back.  So, maybe consider deleting the second one and say, we identified and focused on the following: nutritional support, fluid therapy, oral care, ambulation (delete the mobilization to avoid duplicate) and sputum mobilization or airway clearance and oxygen therapy. 

Thank you for your suggestion. We have deleted the repletion’s on line 174

Thank you for your suggestion of the term ‘airway clearance’ and ‘ambulation’. We have considered the terms and revised the text on line 114, 176, 180, 215 176,

Line #162 – Line #163 – Delete interventions at the beginning of the sentence, just start the sentence with: Oral care, fluid therapy…. were either not performed or were delivered inadequately.  Delete “undone or haphazardly.” 

We have revised the sentence accordingly on line 179-181, at page 5

Line #167 – Delete “that”, not needed.  Here also consider the following revision to clean up and help the flow:  The ethnographic study revealed multiple behavioral factors among RNs, the entire team, management, and the environment responsible for the inadequate and insufficient care. 

Thank you for this suggestion. Revision has been made on line 185-187, at page 5

Line #169 – Line #184 - Consider eliminating extra repetitive sentences and say: Figure 1 presents the primary focus and underlying influence(s) to RNs behaviors limiting their delivery of recommended nursing care.   In addition, figure 2 (additional file) presents the overall overview of behavioral factors and determinants.  Not sure why you have repeated Figure 1 three times.  Just combine the two paragraphs starting line #167 – Line #184.  Then place the figure just after line #184. 

We have revised the paragraph according to your suggestion on line 187-195 and placed the figure after the paragraph.

Line #180 is a repeat from Line #170 reporting on the biomedical aspects limiting nursing care.

We agree and have deleted the sentence on line 204-208

Line #173 – Just abbreviate IDT, not sure why you have the team (interdisciplinary), you have used the abbreviation multiple times.

The sentence has been deleted due to the revision of the paragraph; line 192-194

Line #190 – Change “mobilization” to “ambulation”,

Thank you for your suggestion. We have revised accordingly on line 215 and in rest of the manuscript

Line 200 – Line 220 – You have listed information here and reference table #2.  May consider eliminating some duplication.

Thank you for pointing this out. We have concidered the duplications. We agree that the section ‘3.2.2. Results’ are comprehensive and we have therefore revised the Table 2. We hope that you find the new Table 2 and the revised text on line 225-237 clearer.

Line #228 – Revise to delete extra words: to ensure the appropriate theory-based change methods, we applied the BCTT design or guide.  Delete the extra words at the end.

Thank you for noticing the error. We have noticed this error and have revised the sentence on line 268-269

Line #230 – Line #239 – Not sure if can be simplified for the knowledge information. 

We appreciate your suggestion and have revised the paragraph to simplify the content on line 271-280.  Hopefully for the better.

Line #242 – Change “aims” to “aimed”.

Revised on line 283

Also Line #143 change “includes” to “included” for proper past tense.

We suppose the suggestion is for line 284.

Line #256 – No need to say “we held an event.  Just start the paragraph with: The results of the logic models were presented to the managers and ….

Good suggestion. Revision is made on line 299

Line #257 and Line #258 – Correct “represented” and “influenced”, again past tense.

Agree. Revision has been made on line 299

Line #266 – Delete “below” not needed. 

Revision has been made on line 310

Line #271 – Correct includes to “included”.

Due to previous revision, the term ‘included’ has been deleted. Line 315

Line #277 – Revise to jut in the respiratory unit or ward. Also, delete (before “The.”

Thank you for noticing the typing error. Revision has been made on line 321-322

Line #281 – This sentence is repeat of the

We have concidered your suggestion, and have made several revisions to simplify the content on line 323-329

Line#279 for implementation and execution as a responsibility, please delete.  Or, it sounds like there was an overlap of responsibilities!!!

Due to previous suggestion, we have revised the paragraph and deleted confusing terms. Revision is made on line 323-329

Line #286 – Line #289 -This is a suggestion for you to consider:  The project organization was presented with the logic model of problem (environmental and behavioral) and implementation strategy.  They agreed to the adaptation with allocating resources to deliver the implementation strategy.  Delete (economic and timely resources).   Delete the repeated “the problem” in the parenthesis.

Very good suggestion. Thank you for that. Revision has been made on line 331-337

Line #302 – Line #304 – I think part of the results should be under the Method to describe the plan. Under method consider: The final step was to evaluate the implementation process outcomes such as acceptability, adaptation, … The evaluation plan was developed based on guidance from the stated ….by Proctor et al.

Thank you for your suggestion that we have concidered and tried to accomplish on line 352-363

Line #305 – Line # 320 - Under results now start with: A detailed data collection for the implementation process, outcomes including acceptability, adoption, feasibility, cost, fidelity, and sustainability was developed.  The data collected included: 1) observations……continue until the end of 320 with the except of Line #314 – Delete “on admitted CAP patients” sine you already say electronic patient journals, should this even be patient’s electronic medical records?

We have tried to simplify and revise the paragraph according to the suggestion on line 352-363, 367

DISCUSSION

Line #325 – Delete the second “implementation” and just say consisted on multiple interventions.

We have deleted the term ‘implementation’ and combined the sentences on line 379

Line #332 – Line #338 -What do you mean by “acutely admitted patients???? Consider expanding or clarifying, since later you mention in acute care hospitals missing care is estimated…

Thank you for pointing this out.

CAP is an illness that needs acute diagnostics, treatment, and care. Therefore, most patients are admitted acutely to a hospital. We have expanded the sentence by clarifying what we mean by acutely admitted patients on line 386-390

Maybe starting with: patients admitted to acute care facility are often ill and frail and are provided care in an inpatient setting (physical and psychological).  According to the IM framework, steps I and 2 fundamental care of pneumonia patients are not adequately met.  This is like our study and previous research revealing this as a difficult task.  This is a global phenomenon with an estimated prevalence of 55-98% in acute care hospitals.

We have concidered your suggestion and revised the beginning of the paragraph from line 386-394

Line #338 – Line #339 – Delete the sentence: Low adherence…. does not necessary always cause death.   The care is just not optimal.

We agree on that the low adherence does not cause the death, but it does contribute to the risk of death, but also to the risk of severe morbidity as reported in references 49-55.

Therefore, we would prefer to keep the sentence on line 395

Line #356 – This implementation should be changed to execution.  Sine you just mentioned achieve implementation goal and then you execute the project.

Revised on line 414

Line #362 – Consider changing to: and across the organization. 

Revised on line 421-422

Line #366 – The term powerful or vigorous may be better here instead of “potent”.

Thank you for this suggestion. We have revised the term to ‘effective’. Line 425

Line #391 – Change person to patient-centered care

As described previously, we prefer to keep the term ’person-centered’ on line 451

Line #399 – Change “handled” to “addressed”.

Thank you for this suggestion. We have revised the sentence accordingly on line 459

Line #405 – Not sure if “consumers” may be a better fit than “end-users”.

As the term ‘end-users’ is a commonly used term in an IM approach, and in implementation research in generally, we prefer to keep the term ‘end-users’ on line 465

TABLES

TABLE #1 – I would suggest spelling out IM at the title since we do not have additional abbreviations listed for this table.

Thank you for this suggestion. We have revised the title on line 166

#2 Logic Model of Change – Third bullet under definition, sounds confusing; can you just change it to: Change Objectives: determine the required changes to barriers (as suggested a terminology change previously) to accomplish performance objectives.

Revision has been made to simplify the definition

TABLE #2 – Very difficult to follow, congested and some wording/terminology can be changed.

We agree. As described previously, we have revised the table on line 235-236

Not sure what is the purpose of having “Implementation strategy outcome: Older patients ….in the top row of the table!!! Consider deleting.

We have concidered this suggestion and agree on that it can be deleted

Delete “Abbreviations in table 2; usually the abbreviations are just listed. 

‘Abbreviations in table 2’ is deleted

Under attention may be more appropriate to use: increase awareness on own professional….

Table 2 has been revised to a completely new table.

Under individual level, own professional goal is repeated for all titles, not sure if I understand the purpose.

Table 2 has been revised

TABLE #3 – Same as above: Delete “abbreviations in table 2, just list the abbreviations.

‘Abbreviations in table 3’ is deleted

Under Individual level (first column)

“Education” or “workshops” vs. lectures on professional roles….

We have considered your suggestion, however we prefer to keep ‘lectures’, and have included ‘open debate’ as it was done in the study

Not sure if “Nudging” term is appropriate here.  Have not seen in used in the past for many reviews I have done.  So, maybe change the term.

This is correct, as nudging was used in this study as a theoretically informed method to influence RNs behaviour in delivering EBNC. Nudging method was informed by the Theories of Automatic, Impulsive, and Habitual Behaviour

I believe term encourage will be better for facilitate in the second column; i.e., encourage physicians and or LPNs to support RNs. 

Thank you for this suggestion. We have revised it accordingly

Under environmental Level (third column)

What do you mean by “lectures of the facilitating and eliminating strategies”?

Thank you for this suggestion. As the lecture was designed as a workshop for the management, we have revised the sentence. 

Under Materials (second row)

Delete “Forms”, just say CPW (paper or electronic)

Thank you for this suggestion. We have revised it accordingly

Not sure if you need to be specific about six nursing interventions since

Thank you for this suggestion. We have revised the description and replaced ‘six nursing interventions’ with ‘EBNC interventions’ as described through the manuscript

FIGURE #1.

You have figure on reference on page 5 in three lines #164, 172, and immediately #174). Can this be eliminated and references at just two or even at the end of the section? 

Thank you for this suggestion. We have eliminated the use of ‘Figure 1) on line 184-195

In Environmental determinants –

Not sure if the first bullet point is clear,

We have considered your concern and have revised the point to ‘managerial and environmental determinants’ as this box included also managerial determinants. The first bullet means that, as RNs focused on biomedical aspects of treatment and care, RNs had difficulties to focus on and perform nursing care according to patients’ individual needs and according to evidence-based recommendations for nursing care. Unfortunately, it is a well-known problem in nursing practice.

What do you mean by decentralized management and financialization of hospital services?

In the problem section

Change mobilization to ambulation as mentioned previously.

We agree and have changed the terms through the manuscript.

Non-behavioral factors

What do you mean by the municipality and kitchen’s time schedule…….If it is with regards to nutrition, it may be more appropriate to say get nutritional consult. 

By this sentence we mean that the RNs time schedule was restrained or interrupted by the municipalities visits and the central kitchen’s working schedule, and not the other way around, thus, limiting RNs ability to deliver person- centred EBNC 

Comments on the Quality of English Language

Did not see extensive English Language issues.  However, as mentioned above, I have made several suggestions (including English Language) for the authors for consideration.  The supplemental documents are extensive.  

Thank you. We have made revision thorough the manuscript according to your suggestions and read the manuscript all over again to identify further language errors. Moreover, the manuscript has been revised by the Language Editing Service Scribendi

We agree on that the supplemental documents are extensive. We have thoroughly considered your suggestion and decided to delete the additional file 2 and 3 that can be requested from the corresponding author.

Reviewer 3 Report

Comments and Suggestions for Authors

In this manuscript, the authors describe a systematic process of designing and developing a tailored theory- and research-based implementation strategy based on the Intervention Mapping framework, to support registered nurses in delivering evidence-based and person-centered care for older patients with community acquired pneumonia in a hospital setting. The topic is of great public health importance. The description of strategy development is well articulated and clear, and can be helpful tutorial for readers to help them develop similar strategies tailored to their local needs and feasibility. However, the report would benefit from presenting actual results of the intervention. For example, the authors state that focus groups and interviews were conducted, but the results of these qualitative research methods are missing. 

Comments on the Quality of English Language

n/a

Author Response

For research article: Development of an Implementation Strategy Tailored to Deliver Evidence-Based and Person -Centered Nursing Care for Patients with Community Acquired Pneumonia: An Intervention Mapping Approach.

Manuscript ID: healthcare-2738513

Response to Reviewer 3 Comments

REVIEWER COMMENTS

AUTHORS RESPONSES

Comments and Suggestions for Authors

In this manuscript, the authors describe a systematic process of designing and developing a tailored theory- and research-based implementation strategy based on the Intervention Mapping framework, to support registered nurses in delivering evidence-based and person-centered care for older patients with community acquired pneumonia in a hospital setting. The topic is of great public health importance. The description of strategy development is well articulated and clear, and can be helpful tutorial for readers to help them develop similar strategies tailored to their local needs and feasibility. However, the report would benefit from presenting actual results of the intervention. For example, the authors state that focus groups and interviews were conducted, but the results of these qualitative research methods are missing. 

Thank you very much for taking the time to review our manuscript. We are glad to hear that you find our manuscript important and that you find it as a supportive approach for other experts in a field in solving one of the emerging issues in our healthcare settings globally.  

We have concidered your suggestions thoroughly.  As this manuscript only presents results of the development process, we are not able to present actual results of the intervention. However, the developed strategy has now been tested and evaluated and the results from mentioned focus group interviews, observations etc.  will be submitted followingly. 

We also need to mention that during the implementation strategy development process, several qualitative data were collected. We have presented the results under the section: Step 1, Logic Model of the problem.

We hope our clarification is for your approval.

Please find the corresponding revisions/corrections highlighted/in track changes in the re-submitted files.

Comments on the Quality of English Language

n/a

We have made revision thorough the manuscript and read the manuscript all over again to identify further language errors. Moreover, the manuscript has been revised by the Language Editing Service Scribendi

Round 2

Reviewer 1 Report

Comments and Suggestions for Authors

The authors have greatly improved the article following the indications provided. However, the suggestions relating to the bibliography, which would fit the work very well, were not followed. The authors added references with too much self-citations.

Author Response

For research article: Development of an Implementation Strategy Tailored to Deliver Evidence-Based and Person -Centered Nursing Care for Patients with Community Acquired Pneumonia: An Intervention Mapping Approach.

Manuscript ID: healthcare-2738513

Response to Reviewer 1 Comments

REVIEWER COMMENTS

AUTHORS RESPONSES

Comments and Suggestions for Authors

The authors have greatly improved the article following the indications provided. However, the suggestions relating to the bibliography, which would fit the work very well, were not followed. The authors added references with too much self-citations.

Thank you very much for dedicating your time to thoroughly review our manuscript. We appreciate all your comments and suggestions. We are glad to hear that you see improvements in our manuscript."

We have concidered your further suggestions thoroughly. Please find the detailed responses below and the corresponding revisions/corrections highlighted/in track changes in the re-submitted files.

(1)   Comments regarding the self-citation references

Thank you for conducting a thorough revision of references in our manuscript. We acknowledge that references 12, 19, and 39 are self-citations; nevertheless, they are essential to our study.

Reference 12 is included as it represents our prior work, highlighting variations in in-hospital practices and inconsistencies in the treatment and care of patients with CAP on a national level. This study emphasized the necessity for interventions to optimize in-hospital treatment, with a particular focus on improving nursing care.  However, the reasons for these observed phenomena were not fully understood. Subsequently, we conducted this implementation strategy design study, which has since been tested in a local context.

References 19 and 39 are crucial as they present results from Step 1 of our study, as described on lines 131-178. Moreover, these references are indispensable for a detailed understanding of the implementation strategy design process and illustrate how the findings of the 'needs assessment' guide subsequent steps in Intervention Mapping.

Øverst på formularen

We have thoroughly assessed the inclusion of reference 72, and we concur that it can be omitted. Consequently, we have opted to replace this reference with another citation that we believe makes a more significant contribution to the scholarly content of the paper. Line 632-636

(2)   Recommended references

We have diligently reviewed the recommended references and revisited the content. We concur that the nurse's role in preventing nosocomial infection risk is crucial and should be integrated into their clinical practice. However, it is important to note that our study specifically focuses on patients with community-acquired pneumonia—a condition defined as an infection contracted outside healthcare settings, contrasting with nosocomial infections that occur during the process of receiving healthcare.

Furthermore, our study addresses the design and development of interventions in an implementation strategy aimed at eliminating barriers at the individual, team, management, and environmental levels within a local hospital context. Hence, the study focus is not nursing care interventions (preventive or otherwise), but implementation interventions.

Therefore, we respectfully disagree regarding the utilization of the suggested references in this particular study.

Nevertheless, we genuinely appreciate your feedback, as your suggestion holds significant value and warrants consideration in future studies aimed at optimizing current clinical practices.  We are thus contemplating the integration of your suggestion regarding preventive intervention in our upcoming intervention study within medical departments. This study aims to enhance nursing care, addressing not only patients with community-acquired pneumonia but also extending to all admitted patients who are at risk of nosocomial infection.

Reviewer 3 Report

Comments and Suggestions for Authors

Thank you for clarifying that the intention of this manuscript is to only provide a description of the intervention and to not report any results. I have no further comments. Thank you for your contributions.

Comments on the Quality of English Language

N/A

Author Response

For research article: Development of an Implementation Strategy Tailored to Deliver Evidence-Based and Person -Centered Nursing Care for Patients with Community Acquired Pneumonia: An Intervention Mapping Approach.

Manuscript ID: healthcare-2738513

Response to Reviewer 3 Comments

REVIEWER COMMENTS

AUTHORS RESPONSES

Comments and Suggestions for Authors

Thank you for clarifying that the intention of this manuscript is to only provide a description of the intervention and to not report any results. I have no further comments. Thank you for your contributions.

Thank you for dedicating your time to a thorough review of our manuscript. We are pleased to know that our clarification has been beneficial. Thank you for your valuable contributions.